# Prototypical Prompting for Text-to-image Person Re-identification

## ABSTRACT

In this paper, we study the problem of Text-to-Image Person Re-identification (TIReID), which aims to find images of the same identity described by a text sentence from a pool of candidate images. Benefiting from Vision-Language Pre-training, such as CLIP (Contrastive Language-Image Pretraining), the TIReID techniques have achieved remarkable progress recently. However, most existing methods only focus on instance-level matching and ignore identity-level matching, which involves associating multiple images and texts belonging to the same person. In this paper, we propose a novel prototypical prompting framework (Propot) designed to simultaneously model instance-level and identity-level matching for TIReID. Our Propot transforms the identity-level matching problem into a prototype learning problem, aiming to learn identity-enriched prototypes. Specifically, Propot works by 'initialize, adapt, enrich, then aggregate'. We first use CLIP to generate high-quality initial prototypes. Then, we propose a domain-conditional prototypical prompting (DPP) module to adapt the prototypes to the TIReID task using task-related information. Further, we propose an instance-conditional prototypical prompting (IPP) module to update prototypes conditioned on intra-modal and inter-modal instances to ensure prototype diversity. Finally, we design an adaptive prototype aggregation module to aggregate these prototypes, generating final identity-enriched prototypes. With identity-enriched prototypes, we diffuse its rich identity information to instances through prototype-to-instance contrastive loss to facilitate identity-level matching. Extensive experiments conducted on three benchmarks demonstrate the superiority of Propot compared to existing TIReID methods.

## CCS CONCEPTS

• **Computing methodologies** → **Visual content-based indexing and retrieval**.

## KEYWORDS

Text-to-image person re-identification, Identity-level matching, Prototypical prompting

## 1 INTRODUCTION

Person re-identification (ReID), devoted to searching a person-of-interest across different times, locations, and camera views, has

*ACM MM, 2024, Melbourne, Australia*
© 2024 Copyright held by the owner/author(s). Publication rights licensed to ACM.
ACM ISBN 978-x-xxxx-xxxx-x/YY/MM
https://doi.org/10.1145/nnnnnnn.nnnnnnn

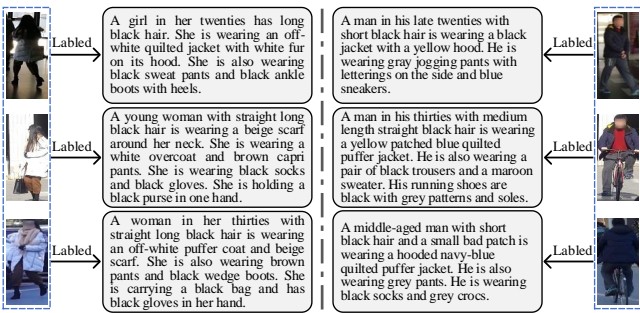

(a) Example of TIReID data

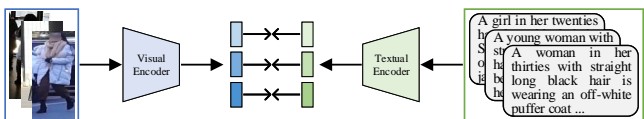

(b) Existing matching paradigm

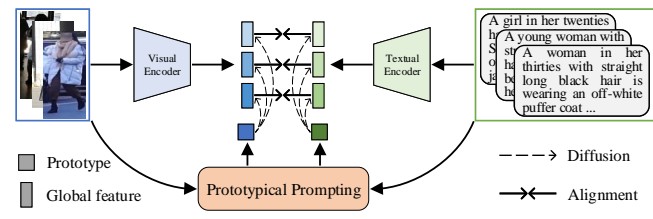

Prototype · Global feature · Prototypical Prompting · Diffusion · Alignment

(c) Our proposed Propot

**Figure 1: The motivation of our proposed Propot. (a) In TIReID data, Instances under the same identity show significant differences. (b) Most existing TIReID methods only focus on instance-level matching and ignore identity-level matching. (c) Our Propot proposes a prototype prompting framework to create identity-enriched prototypes and diffuse their rich identity information to instances for modeling identity-level matching.**

garnered increasing interest due to its huge practical value. In recent years, we have witnessed great progress on image-to-image person re-identification (IIReID) [24, 28, 50, 58, 61], which has been successfully applied in various practical scenarios. However, ReID is still challenging when pedestrian images from some cameras are missing. In contrast, textual descriptions are more readily accessible and freer than images collected by specialized equipment, which can be obtained from witnesses at the scene. Thus, text-to-image person re-identification (TIReID) [30] has received a lot of research attention recently owing to it being closer to real-world scenarios.

Compared to IIReID, TIReID faces a significant challenge in bridging the modal gap between images and texts while modeling their correspondence. Various methods have been devised to

address this challenge by cross-modal interactive attention mechanisms [39, 63, 66], semantically aligned local feature learning [5, 8, 41, 57], and fine-grained auxiliary task-enhanced global feature learning [37, 38]. Nowadays, the emergence of vision-language pre-training models (VLP) has propelled the advancements in computer vision, showcasing exceptional capabilities in semantic understanding, multi-modal alignment, and generalization. It is intuitive to consider that the rich multi-modal prior information of VLP can be harnessed to enhance TIReID models in modeling inter-modal correspondence. Some follow-up CFine [56], IRRA [20] extend VLP for TIReID, greatly promoting the progress of this task.

Despite the success of VLP-based methods, the performance of TIReID still lags behind that of IIReID, especially in complex scenes like multiple cameras and drastic view changes. This is primarily due to the specificity of the TIReID task. As shown in Figure 1(a), there are multiple images of the same identity, each annotated with specific text at the instance level. The goal of TIReID is to match all images of the same identity given a text, meaning to correctly associate each text with all images of the same identity. However, most existing methods, constrained by the alignment objective function [55], only consider **instance-level matching** between each image and its annotated text, ignoring matches with other non-annotated texts of the same identity, called **identity-level matching**. An intuitive idea to this problem is to directly match all images and texts under the same identity, but this can disrupt instance-level matching relationship [55] due to huge view variations, leading to performance collapse. So far, there have been limited solutions [8, 55] proposed to model identity-level matching for TIReID. SSAN [8] introduces an enhanced alignment loss to simultaneously align the image with its annotated text and another text sharing the same identity. LCR$^2$S [55] designs a teacher-student network architecture to randomly fuse two images/texts under the same identity and then align them to model identity-level matching. While effective, they do not comprehensively model the matching relationship between all images and texts under the same identity. Moreover, the two-stage teacher-student framework is costly to train and lacks practicality. In this paper, we thus ask: *Can an end-to-end efficient TIReID model be trained to simultaneously model instance-level and identity-level matching?*

To address this problem, we propose a novel **Pro**totypical **prompting** framework (**Propot**) that enables the network to model instance-level and identity-level matching for TIReID simultaneously. As shown in Figure 1, existing TIReID methods only model instance-level matching between each image and its labeled text. In contrast, our Propot further introduces identity-level matching based on instance-level matching. We learn a prototype containing rich identity information for each identity and diffuse the rich information from the prototype to each instance, indirectly modeling identity-level matching. This transforms the identity-level matching problem into an **identity-enriched prototype learning** problem.

Specifically, our Propot follows the 'initialize, adapt, enrich, then aggregate' pipeline. Initially, we leverage CLIP's strong multi-modal alignment capability to cluster instance (image/text) features under each identity, forming the initial prototype for each identity. Since there's a gap between CLIP and TIReID data, this prototype is not fully adapted to TIReID. To address this, we propose a domain-conditional prototypical prompting (DPP) module inspired

by CoOp [65], which introduces a set of learnable prompt tokens to learn target domain knowledge and adapt the initial prototype to the TIReID task. However, instances under the same identity exhibit significant diversity due to factors like view changes and camera parameters. Ignoring this diversity can lead to a monotonous prototype, losing rich identity information. Thus, inspired by CoCoOp [46], we propose an instance-condition prototypical prompting (IPP) module. This module generates two prototypical prompts conditioned on a batch of instances, leveraging both intra-modal and inter-modal instances to enhance diversity and bridge the modal gap. To integrate the multiple prototypes generated above, an adaptive prototype aggregation (APA) module is designed, which treats the initial CLIP-generated prototype as the baseline and adaptively ensemble these generated prototypes as the final prototype. Finally, we utilize a prototype-to-instance contrastive loss to diffuse the rich identity information from the prototype to each instance, enabling effective modeling of identity-level matching. Our Propot is single-stage and end-to-end trainable. During inference, only the backbone of the network is used for inference, which is simple and efficient.

Here are the main contributions of our paper: (1) We propose an end-to-end trainable prototypical prompting framework to model instance-level and identity-level cross-modal matching for TIReID simultaneously. (2) We transform the identity-level matching problem into an identity-enriched prototype learning problem. We use CLIP to generate initial prototypes and propose a domain-conditional prototypical prompting (DPP) module and an instance-condition prototypical prompting (IPP) module to generate multiple identity-enriched prototypes. An adaptive prototype aggregation (APA) module is designed to fully integrate these prototypes. (3) Extensive experiments have been conducted to validate the effectiveness of Propot, and it achieves superior performance on the CUHK-PEDES, ICFG-PEDES, and RSTPReid benchmarks.

## 2 RELATED WORK

### 2.1 Text-to-Image Person Re-identification

TIReID [30] has gained significant attention in recent years. Existing studies in TIReID can be broadly divided into the following categories: better model architectures and optimization losses, better alignment strategies, and richer prior information. Previous methods [3, 62, 64] have focused on designing network and optimization loss to learn globally aligned image and text features in a joint embedding space. These methods are simple and efficient, but ignore detailed information and fine-grained correspondences. To address these limitations, subsequent methods have refined matching strategies to mine fine-grained correspondence between modalities. Some methods [22, 30, 39, 47, 63, 66] have emerged to achieve fine-grained matching through interactions between local parts of images and texts. While effective, these approaches require significant computational resources. To mitigate computational overhead, other works [5, 8, 53] adopt local image parts as references to guide the generation of locally aligned text features, avoiding pairwise interactions. However, the effectiveness of these methods depends on the quality of explicitly acquired local parts. As an alternative, diverse aggregation schemes [27, 41, 44, 57] have been proposed to

adaptively aggregate images and text into modality-shared local features, avoiding explicit local part acquisition.

Recently, visual-language pre-training models (VLP) [40] have made significant progress. To leverage their rich multi-modal knowledge, recent advanced methods [2, 15, 20, 56] have proposed diverse strategies to tailor VLP for TIReID, resulting in notable performance enhancements. However, despite these advancements, existing methods often overlook the specific nature of TIReID as a challenge involving matching multi-view image-text pairs at the identity level, rather than merely at the instance level. Addressing this concern, LCR$^2$S [55] devised a teacher-student network to reason about a comprehensive representation with multi-view information from a single image or text. This approach yielded notable performance gains, it had limitations in fully integrating multi-view information and exhibited lower training efficiency. In this study, we present an end-to-end framework to learn a prototype with comprehensive information for each identity using the training set, and transfer multi-view information to individual samples for identity-level matching. This approach enhances both efficiency and the integration of multi-view data for TIReID.

## 2.2 Vision-Language Pre-Training

Nowadays, the "pre-training and fine-tuning" paradigm stands as a foundational approach in the computer vision community, in which pre-training models that can provide rich prior knowledge for various downstream vision tasks have gained increasing attention. Previous prevailing practice is rooted in the supervised unimodal pre-training [9, 16] on ImageNet [6]. However, to improve representation capabilities and overcome annotation constraints, a new paradigm called language-supervised vision pre-training (vision-language pre-training, VLP) has emerged. Within VLP, investigating the interaction between vision and language has become a central research focus. Several works [4, 25, 26, 33, 49] have been proposed to model the interaction of vision and language based on some multi-modal reasoning tasks, such as masked language/region modeling, and image captioning. Recently, contrastive representation learning [19, 40, 59] has gained attention, which learns representations by contrasting positive pairs against negative pairs. The representative work, Contrastive Language-Image Pretraining (CLIP) [40], has strong multi-modal semantic representation and zero-shot generalization capabilities, which is trained on 400 million image-text pairs. CLIP has shown promising adaptability for various downstream tasks like video-text retrieval [10, 34], referring image segmentation [51], and person re-identification [56]. Our work builds upon CLIP and utilizes its ample multi-modal knowledge to learn identity-enriched prototypes.

## 2.3 Prompt Learning

Prompt learning [21, 43], originating from natural language processing (NLP), is a method used to customize pre-training models for different tasks by providing instructions in the form of sentences, known as prompts. Early prompts were manually designed for specific tasks, but recent studies have introduced prompt learning, where task-specific prompts are automatically generated during fine-tuning. This approach addresses issues like instability and knowledge bias. Prompt learning has now been extended to computer vision. CoOp [65] pioneered the application of prompt

learning to adapt large vision-language models in computer vision, while CoCoOp [46] built upon CoOp to introduce a conditional prompt learning framework, improving generalization. Chen et al. [23] proposed efficiently adapting the CLIP model to the video understanding task by optimizing a few continuous prompt vectors. CLIP-ReID [29] designed a two-stage framework to generate coarse descriptions of pedestrians, leveraging CLIP's capabilities for ReID. Inspired by these, in this work we propose using prompt learning to learn comprehensive prototype representations to model identity-level matching for TIReID.

## 3 THE PROPOT FRAMEWORK

Our Propot is a conceptually simple end-to-end trainable framework, and the overview is depicted in Figure 2. We first extract features of images and texts through visual and textual encoders, followed by instance-level and identity-level matching.

## 3.1 Feature Extraction

Previous studies [2, 20, 56] have underscored the effectiveness of CLIP [40] in tackling TIReID challenges. To harness the vast multi-modal knowledge in CLIP, we leverage its image and text encoders to initialize Propot's image and text backbones. Concretely, for an image-text pair $(I, T)$, we exploit CLIP pre-trained ViT model to extract visual representations for the image $I$, resulting in the global visual feature $v \in \mathbb{R}^d$. For the text caption $T$, the CLIP's textual encoder is utilized to generate the global textual feature $t \in \mathbb{R}^d$.

Instance-level matching involves directly aligning $v$ with $t$. For identity-level matching, we transform it into an identity-enriched prototype learning problem. The aim is to learn a rich prototype containing all instance identity information for each identity in the training set. In Propot, the prototype learning method is crucial. While it is conceivable to learn a prototype for each identity from scratch, akin to previous method [29], such an approach can encounter challenges in network convergence and may not ensure prototype quality. To address these concerns, we introduce a novel 'initialize, adapt, enrich, then aggregate' prototype learning scheme, detailed in the subsequent subsections.

## 3.2 Initial Prototype Generation

We start our approach by utilizing the CLIP model to extract features for each instance in the TIReID training set, leveraging its strong semantic information extraction and multi-modal alignment capabilities. We then cluster these instance features based on shared identity labels to produce initial prototypes for each modality. Specifically, given the training set $\{I_i, T_i, Y_i\}_{i=1}^{N_s}$, where $Y_i \in \{L_1, L_2, ..., L_N\}$ represents the identity label of the image-text pair $(I_i, T_i)$, $N_s$ denotes the number of pairs, and $N$ denotes the number of identity label, we employ the pre-trained CLIP visual and textual encoders to obtain the visual and textual features $\{v_i, t_i\}_{i=1}^{N_s}$ of all image-text pairs. We then perform feature clusters on the image and text features. Taking identity $L_i$ as an example, We generate initial prototypes $pt_i^v$ and $pt_i^t$ for identity $L_i$ as follows:

$$pt_i^v = \sum_{j=1, Y_j \in L_i}^{N_i} v_j, \quad pt_i^t = \sum_{j=1, Y_j \in L_i}^{N_i} t_j, \quad (1)$$

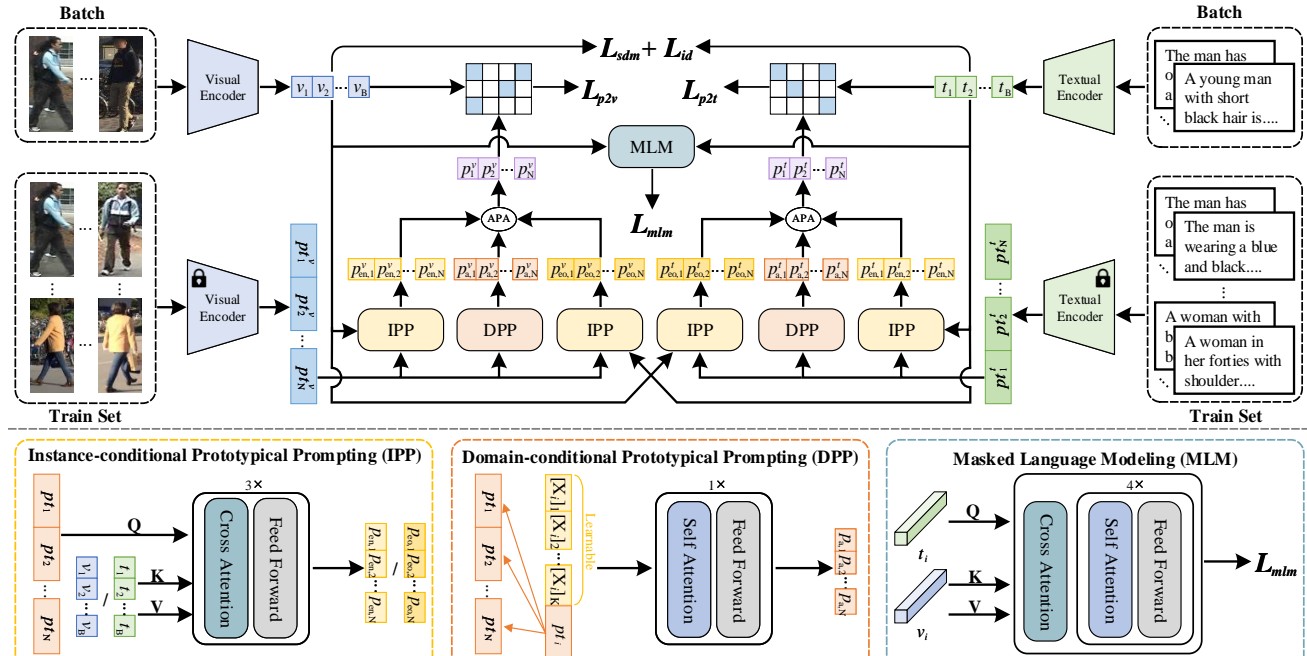

**Figure 2: Overview of our Propot. It includes instance-level matching and identity-enriched prototype learning. For instance-level matching, each image and its annotated text are directly aligned through SDM loss (Baseline). For prototype learning, we first utilize pre-trained CLIP to generate initial prototypes ($pt^v$ and $pt^t$). We then adapt the initial prototypes to TIReID through the DPP module, resulting in task-adapted prototypes ($p_a^v$ and $p_a^t$). The IPP module updates the prototypes conditioned on a batch of intra-modal and inter-modal instances, generating intra-modal and inter-modal enriched prototypes ($p_{en}^v$, $p_{en}^t$, $p_{eo}^v$ and $p_{eo}^t$). The multiple prototypes are aggregated using Adaptive Prototypical Aggregation (APA) to generate final prototypes ($p^v$ and $p^t$). Their rich identity information is then diffused to each instance using prototype-to-instance contrastive loss ($\mathcal{L}_{p2v}$, $\mathcal{L}_{p2t}$) to model identity-level matching. Moreover, we introduce the MLM module to enhance fine-grained matching. During testing, only visual and textual encoders are used for inference.**

where $N_i$ denotes the number of instances under the identity $L_i$. Therefore, for the entire training set, we can generate visual and textual initial prototype sets $pt^v = [pt_1^v, pt_2^v, ..., pt_N^v] \in \mathbb{R}^{N \times d}$ and $pt^t = [pt_1^t, pt_2^t, ..., pt_N^t] \in \mathbb{R}^{N \times d}$.

## 3.3 Domain-conditional Prototypical Prompting

Benefiting from CLIP's powerful capabilities, the initial prototype effectively captures some identity information to model identity-level matching, leading to improved performance (as shown in Table 4), which confirms the validity of our idea. However, a notable challenge arises due to the domain gap between the pre-training data of CLIP and the TIReID dataset. Consequently, the identity information mined by CLIP's features falls short, significantly diminishing the impact of the initial prototype. To address this challenge, inspired by the Contextual Optimization (CoOp) framework [65], which introduces learnable context vectors to adapt VLP to downstream tasks, we introduce a Domain-conditional Prototypical Prompting (DPP) module to adjust the initial prototype to the TIReID task. Specifically, we add a set of learnable contextual prompt vectors before each initial prototype. These vectors undergo training on the TIReID dataset concurrently with the network, gaining domain knowledge specific to the TIReID task. These vectors then transmit the domain knowledge to each initial

prototype through the self-attention encoder (SAE), aiding in the adaptation of prototypes to the TIReID task.

Formally, for each initial prototype $pt_i$, we add a set of learnable contextual prompt vectors $\{[X_i]_1, [X_i]_2, ..., [X_i]_K\} \in \mathbb{R}^{K \times d}$ before it, where $[X_i]$ is the visual contextual prompt vector $c_{v,i}$ for the visual prototype $pt_i^v$, and $[X_i]$ is the textual contextual prompt vector $c_{t,i}$ for the textual prototype $pt_i^t$. Then, we feed $\{[X_i]_1, [X_i]_2, ..., [X_i]_K, pt_i\} \in \mathbb{R}^{(K+1) \times d}$ into SAE to pass the information to the initial prototype for updating it.

$$p_{a,i} = SAE(\{[X_i]_1, [X_i]_2, ..., [X_i]_K, pt_i\}), \quad (2)$$

where $p_{a,i} \in \mathbb{R}^d$ represents the task-adaptive prototype. SAE is comprised of $N_a$ blocks, with each block containing a multi-head self-attention layer and a feed-forward network layer. The above process allows us to generate modality-specific task-adaptive prototypes, $p_{a,i}^v$ and $p_{a,i}^t$, for the input prototypes of different modalities.

## 3.4 Instance-conditional Prototypical Prompting

So far, we have generated a prototype adapted to the TIReID task. As expected, the resulting prototype brought significant performance improvements. However, as depicted in Figure 1, instances of the same identity showcase notable diversity due to factors like view

changes and camera parameters. The previously derived prototype fails to account for this diversity, resulting in the loss of some discriminative identity information. To solve this problem, inspired by Conditional Context Optimization (CoCoOp) [46], we propose an Instance-conditional Prototypical Prompting (IPP) module. This module updates the prototype conditioned on each batch of the input instances, enabling the comprehensive mining of instance-specific information and ensuring diversity within the prototype.

Specifically, for a batch of $B$ image-text pairs $\{I_i, T_i\}_{i=1}^{B}$, after feature encoding, we have $V_B = [v_1, v_2, ..., v_B] \in \mathbb{R}^{B \times d}$ and $T_B = [t_1, t_2, ..., t_B] \in \mathbb{R}^{B \times d}$. Next, the instance features $V_B/T_B$ and the prototypes $pt^v/pt^t$ are processed in a cross-attention decoder (CAD). This decoder enables interaction between instance features and prototypes, extracting identity-relevant information from instances to enrich the prototypes. There are $N_e$ blocks in CAD, each featuring a multi-head cross-attention layer and a feed-forward network layer. For the cross-attention decoder, the prototypes $pt^v/pt^t$ serve as *query*, while the instance features $V_B/T_B$ are designated as *key* and *value*, which is formulated as follows:

$$p_{en,i}^v = CAD(pt^v, V_B, V_B), \quad p_{en,i}^t = CAD(pt^t, T_B, T_B), \quad (3)$$

where $p_{en,i}^v \in \mathbb{R}^d$ and $p_{en,i}^t \in \mathbb{R}^d$ represent the intra-modal instance-enriched visual and textual prototypes, respectively. This process enhances the prototypes using intra-modal instance features. Additionally, we extend this enrichment by incorporating inter-modal instance features, fostering the network to extract modality-shared identity information comprehensively and minimize the modal gap. We formulate them as

$$p_{eo,i}^v = CAD(pt^v, T_B, T_B), \quad p_{eo,i}^t = CAD(pt^t, V_B, V_B), \quad (4)$$

where $p_{eo,i}^v \in \mathbb{R}^d$ and $p_{eo,i}^t \in \mathbb{R}^d$ represent the inter-modal instance-enriched visual and textual prototypes, respectively.

## 3.5 Adaptive Prototype Aggregation

For both images and texts, we generate three distinct modality-specific prototypes: $p_{a,i}^m \in \mathbb{R}^d$, $p_{en,i}^m \in \mathbb{R}^d$, and $p_{eo,i}^m \in \mathbb{R}^d$, each containing different information for each identity $L_i$, where $m \in \{v, t\}$. To seamlessly integrate all available information, we introduce an Adaptive Prototype Aggregation (APA) module. This module effectively aggregates diverse prototypes to form a comprehensive identity-enriched prototype. Since $pt_i^m$ is directly derived by CLIP, ensuring high-quality prototypes due to its robust semantic understanding capabilities. We designate $pt_i^m$ as the prototype baseline for aggregation. The aggregation weights are determined based on the correlation of other prototypes with $pt_i^m$. This approach enables the suppression of spurious identity prototype information in $p_{a,i}^m$, $p_{en,i}^m$, $p_{eo,i}^m$, while amplifying the correct ones during aggregation. The calculation of prototype correlation, serving as the aggregation weights for the three prototypes, is as follows:

$$w_{a,i}^m = pt_i^m(p_{a,i}^m)^T, \quad w_{en,i}^m = pt_i^m(p_{en,i}^m)^T, \quad w_{eo,i}^m = pt_i^m(p_{eo,i}^m)^T. \quad (5)$$

Then, we utilize the softmax function to normalize the weights and obtain the final identity-enriched prototype as

$$p_i^m = pt_i^m + \sum_k w_{k,i}^m \cdot p_{k,i}^m, \quad (6)$$

where $k \in \{a, en, eo\}$. Thus, for the entire training set, we can generate the final visual and textual prototype sets $p^v = [p_1^v, p_2^v, ..., p_N^v] \in \mathbb{R}^{N \times d}$ and $p^t = [p_1^t, p_2^t, ..., p_N^t] \in \mathbb{R}^{N \times d}$. Subsequently, we propagate the rich identity information encapsulated in the prototypes to each instance through prototype-to-instance contrastive loss to model identity-level matching for TIReID.

## 3.6 Training and Inference

The goal of Propot is to model both instance-level and identity-level matching for TIReID. To this end, we optimize Propot through cross-modal matching loss, cross-entropy loss, prototype-to-instance contrastive loss, and mask language modeling loss.

Given a batch of $B$ image-text pairs $\{I_i, T_i\}_{i=1}^{B}$, we generate the global visual and textual features as $V_B = [v_1, v_2, ..., v_B] \in \mathbb{R}^{B \times d}$ and $T_B = [t_1, t_2, ..., t_B] \in \mathbb{R}^{B \times d}$. To align each image $I_i$ and its annotated text $T_i$, we utilize the similarity distribution matching (SDM) [20] as the cross-modal matching loss to model instance-level matching between them.

$$\mathcal{L}_{sdm} = \mathcal{L}_{i2t} + \mathcal{L}_{t2i}, \quad (7)$$

$$\mathcal{L}_{i2t} = \frac{1}{B} \sum_{i=1}^{B} \sum_{j=1}^{B} p_{i,j} log(\frac{p_{i,j}}{q_{i,j} + \epsilon}), \quad (8)$$

$$p_{i,j} = \frac{exp(sim(v_i, t_j)/\tau)}{\sum_{k=1}^{B} exp(sim(v_i, t_k)/\tau)}, \quad (9)$$

where $sim(\cdot)$ denotes the cosine similarity function, $\tau$ denotes the temperature factor to control the distribution peaks, and $\epsilon$ is a small number to avoid numerical problems. $q_{i,j}$ denotes the true matching probability. $\alpha$ indicates the margin. $\mathcal{L}_{t2i}$ can be obtained by exchanging $v$ and $t$ in Eqs. 8 and 9. Moreover, to ensure the discriminability of features $v$ and $t$, we calculate cross-entropy loss $\mathcal{L}_{id}$ on them to classify them into corresponding identity labels.

Through the prototype learning process described above, we generate two identity-rich prototypes $p_i^v$ and $p_i^t$ for each identity $L_i$. To effectively diffuse the rich identity information encapsulated in the prototypes to instances of the same identity, we employ a prototype-to-instance contrastive loss, denoted as $\mathcal{L}_{p2i}$. This loss operates in tandem with the cross-modal matching loss, collectively contributing to the modeling of identity-level matching.

$$\mathcal{L}_{p2i} = \sum_{i=1}^{N} \mathcal{L}_{p2v}(L_i) + \mathcal{L}_{p2t}(L_i), \quad (10)$$

$$\mathcal{L}_{p2v}(L_i) = -\frac{1}{|P(L_i)|} \sum_{p \in P(L_i)} \frac{exp(sim(v_p, p_i^v)/\tau)}{\sum_{k=1}^{B} exp(sim(v_j, p_i^v)/\tau)}, \quad (11)$$

$$\mathcal{L}_{p2t}(L_i) = -\frac{1}{|P(L_i)|} \sum_{p \in P(L_i)} \frac{exp(sim(t_p, p_i^t)/\tau)}{\sum_{k=1}^{B} exp(sim(t_j, p_i^t)/\tau)}, \quad (12)$$

where $P(L_i)$ represents a set of instance indices of identity $L_i$, and $|P(L_i)|$ denotes the cardinality of $P(L_i)$. To further improve performance, we follow [20] to introduce a mask language modeling task $\mathcal{L}_{mlm}$ to model fine-grained matching between modalities.

Propot is a single-stage and end-to-end trainable framework, and the overall objective function $\mathcal{L}$ for training is as follows:

$$\mathcal{L} = \mathcal{L}_{sdm} + \mathcal{L}_{id} + \lambda_1 \mathcal{L}_{p2i} + \lambda_2 \mathcal{L}_{mlm}, \quad (13)$$

where $\lambda_1$ and $\lambda_2$ balance the contribution of different loss terms.

The prototype learning process only exists during training. During inference, we only use Propot's visual and textual encoders to extract the global features of the test samples. The retrieval results are obtained by calculating the cosine similarity.

## 4 EXPERIMENTS

### 4.1 Experiment Settings

**Datasets and Metrics:** The evaluations are conducted on three datasets for TIReID. **CUHK-PEDES** [30] has 40,206 images and 80,412 descriptions of 13,003 persons. Each image has 2 descriptions, each with an average length of 23 words. The training set has 34,054 images and 68,108 descriptions of 11,003 persons, the validation set includes 3,078 images and 6,156 descriptions of 1,000 persons, and the testing set involves 3,074 images and 6,148 descriptions of 1,000 persons. **ICFG-PEDES** [8] comprises 54,522 image-text pairs of 4,102 persons, with descriptions averaging 37 words in length. We utilize 34,674 pairs of 3,102 persons for training and reserving the remaining 1,000 persons for evaluation. **RSTPReid** [66] includes 20,505 images of 4,101 persons, each annotated with 2 descriptions, with descriptions averaging 23 words in length. There are 3,701 persons in the training set, 200 persons in the validation set, and 200 persons in the testing set. For performance evaluation, we employ the Rank-$k$ matching accuracy (R@$k$, $k$=1, 5, 10).

**Implementation Details:** For input images, we uniformly resize them to 384×128 and augment them with random horizontal flipping, random crop with padding, and random erasing. For input texts, the length of the token sequence is unified to 77 and augmented by randomly masking out some tokens with a 15% probability [7]. Propot's visual encoder is initialized with the CLIP-ViT-B/16 version of CLIP, where the dimension $d$ of global visual and textual features is 512. The SAE and CAD consist of $N_a = 1$ and $N_e = 3$ blocks, respectively, each with 8 heads. The length $K$ of the learnable prompt vector in the DPP module is set to 4. The temperature factor $\tau$ is 0.02. The loss balance factors are set to $\lambda_1 = 0.2$ and $\lambda_2 = 1.0$. Model training utilizes the Adam optimizer with a weight decay factor of 4e-5. Initial learning rates are 1e-5 for the visual/textual encoder and 1e-4 for other network modules. We employ a cosine learning rate decay strategy, stopping training at 60 epochs. The learning rate linearly decays by a factor of 0.1 within the first 10% of the training epochs for warmup. All experiments are implemented in PyTorch library, and models are trained with a batch size of 64 on a single RTX3090 24GB GPU.

### 4.2 Comparisons with State-of-the-art Models

In this section, we compare our Propot with current state-of-the-art (SOTA) approaches on all three TIReID benchmarks. The methods for comparison are categorized into two sections: methods (w/o CLIP) based on single-modal pre-training models (ResNet [16], ViT [9], BERT [7]) and methods (w/ CLIP) based on multi-modal pre-training CLIP [40] models. Propot falls under the second section.

The performance comparison with SOTA methods on **CUHK-PEDES** is summarized in Table 1. The proposed Propot framework demonstrates competitive performance at all metrics and outperforms all compared methods except [31], achieving remarkable R@1, R@5, and R@10 accuracies of 74.89%, 89.90% and 94.17%, respectively. While our R@1 accuracy is slightly lower (-0.13%)

**Table 1: Performance comparison with state-of-the-art methods on CUHK-PEDES. R@1, R@5, and R@10 are listed.**

| Methods | Pre | Ref | R@1 | R@5 | R@10 |
|---|---|---|---|---|---|
| SRCF [45] | | ECCV'22 | 64.04 | 82.99 | 88.81 |
| LBUL [52] | | MM'22 | 64.04 | 82.66 | 87.22 |
| AXM-Net [11] | | AAAI'22 | 64.44 | 80.52 | 86.77 |
| $C_2A_2$ [37] | | MM'22 | 64.82 | 83.54 | 89.77 |
| LGUR [41] | | MM'22 | 65.25 | 83.12 | 89.00 |
| FedSH [35] | w/o CLIP | TMM'23 | 60.87 | 80.82 | 87.61 |
| PBSL [42] | | MM'23 | 65.32 | 83.81 | 89.26 |
| BEAT [36] | | MM'23 | 65.61 | 83.45 | 89.57 |
| MANet [57] | | TNNLS'23 | 65.64 | 83.01 | 88.78 |
| ASAMN [38] | | TIP'23 | 65.66 | 84.53 | 90.21 |
| $LCR^2S$ [55] | | MM'23 | 67.36 | 84.19 | 89.62 |
| TransTPS [1] | | TMM'23 | 68.23 | 86.37 | 91.65 |
| MGCN [14] | | TMM'23 | 69.40 | 87.07 | 90.82 |
| CFine [56] | | TIP'23 | 69.57 | 85.93 | 91.15 |
| VLP-TPS [48] | | arXiv'23 | 70.16 | 86.10 | 90.98 |
| VGSG [17] | | TIP'23 | 71.38 | 86.75 | 91.86 |
| IRRA [20] | | CVPR'23 | 73.38 | 89.93 | 93.71 |
| BiLMa [12] | | ICCVW'23 | 74.03 | 89.59 | 93.62 |
| TCB [60] | | MM'23 | 74.45 | 90.07 | 94.66 |
| DCEL [31] | w/ CLIP | MM'23 | 75.02 | 90.89 | 94.52 |
| SAL [13] | | MMM'24 | 69.14 | 85.90 | 90.81 |
| EESSO [54] | | IVC'24 | 69.57 | 85.65 | 90.71 |
| PD [32] | | arXiv'24 | 71.59 | 87.95 | 92.45 |
| CFAM [67] | | CVPR'24 | 72.87 | 88.61 | 92.87 |
| TBPS-CLIP [2] | | AAAI'24 | 73.54 | 88.19 | 92.35 |
| **Ours** | | - | **74.89** | **89.90** | **94.17** |

compared to the optimal method DCEL [31], it is essential to note that DECL introduces both mask language modeling and global-local semantic alignment to mine fine-grained matching, resulting in higher computational cost. In contrast, Propot employs only mask language modeling for fine-grained matching. Additionally, as observed in Table 4#7, even without a local matching module, Propot achieves a noteworthy 74.37% R@1 accuracy, surpassing most compared methods. Table 2 reports the comparative results on **ICFG-PEDES**. Our Propot establishes a new SOTA performance on this dataset, with R@1, R@5, and R@10 accuracy scores of 65.12%, 81.57%, and 86.97%, respectively. Notably, Propot surpasses the current SOTA solution TBPS-CLIP [2] by 1.23% in R@5 and 1.50% in R@10. The comparative analysis with SOTA methods on **RSTPReid** is summarized in Table 3. Propot demonstrates commendable performance, achieving competitive results over recent SOTA methods, specifically attaining 61.83%, 83.45%, and 89.70% on R@1, R@5, and R@10. Although our method achieves slightly lower performance (-0.08%) than the optimal method TBPS-CLIP [2], which incorporates CLIP into the TIReID task using various data augmentation and training tricks. In contrast, our approach uses only basic data augmentation without additional tricks. In summary, propot achieves superior performance on all three benchmarks. This is attributed to the fact that our prototypical prompting can simultaneously model instance-level and identity-level matching.

### 4.3 Ablation Studies

We conduct ablation experiments for Propot on CUHK-PEDES using the default settings above. Baseline solely includes visual and textual encoders initialized by CLIP, which is trained using SDM and cross-entropy loss, with the training settings aligned with those of Propot.

**Table 2: Performance comparison with state-of-the-art methods on ICFG-PEDES. R@1, R@5, and R@10 are listed.**

| Methods | Pre | Ref | R@1 | R@5 | R@10 |
|---|---|---|---|---|---|
| IVT [44] | | ECCVW'22 | 56.04 | 73.60 | 80.22 |
| SRCF [45] | | ECCV'22 | 57.18 | 75.01 | 81.49 |
| LGUR [41] | | MM'22 | 57.42 | 74.97 | 81.45 |
| FedSH [35] | | TMM'23 | 55.01 | 72.75 | 79.48 |
| ASAMN [38] | w/o CLIP | TIP'23 | 57.09 | 76.33 | 82.84 |
| PBSL [42] | | MM'23 | 57.84 | 75.46 | 82.15 |
| LCR$^2$S [55] | | MM'23 | 57.93 | 76.08 | 82.40 |
| BEAT [36] | | MM'23 | 58.25 | 75.92 | 81.96 |
| MANet [57] | | TNNLS'23 | 59.44 | 76.80 | 82.75 |
| MGCN [14] | | TMM'23 | 60.20 | 76.75 | 83.90 |
| VLP-TPS [48] | | arXiv'23 | 60.64 | 75.97 | 81.76 |
| CFine [56] | | TIP'23 | 60.83 | 76.55 | 82.42 |
| TCB [60] | | MM'23 | 61.60 | 76.33 | 81.90 |
| VGSG [17] | | TIP'23 | 63.05 | 78.43 | 84.36 |
| IRRA [20] | | CVPR'23 | 63.46 | 80.25 | 85.82 |
| BiLMa [12] | | ICCVW'23 | 63.83 | 80.15 | 85.74 |
| DCEL [31] | w/ CLIP | MM'23 | 64.88 | 81.34 | 86.72 |
| EESSO [54] | | IVC'24 | 60.84 | 77.89 | 83.53 |
| PD [32] | | arXiv'24 | 60.93 | 77.96 | 84.11 |
| CFAM [67] | | CVPR'24 | 62.17 | 79.57 | 85.32 |
| SAL [13] | | MMM'24 | 62.77 | 78.64 | 84.21 |
| TBPS-CLIP [2] | | AAAI'24 | 65.05 | 80.34 | 85.47 |
| **Ours** | | - | **65.12** | **81.57** | **86.97** |

**Contributions of Proposed Components:** In Table 4, we assess the contribution of each module of Propot, including the initial prototype (IniPt) generated by CLIP, the DPP module, the IPP module, and the MLM module. Baseline achieves a notable R@1 accuracy of 72.73% due to the rich multi-modal knowledge of CLIP. Introducing our prototype learning process based on Baseline, aimed at modeling identity-level matching, yields several key observations. Firstly, using only the initial prototype generated by CLIP to supervise identity-level matching leads to distinct performance improvements (+0.35% R@1 improvement over Baseline), affirming the feasibility of our approach and the importance of identity-level matching. Secondly, incorporating the DPP module to update the initial prototype results in a 1.38% R@1 accuracy improvement over Baseline, demonstrating the effective adaptation of the initial prototype to the TIReID task. Thirdly, when the IPP module is employed to update the prototype, whether conditioned on intra-modal or inter-modal instances, substantial performance enhancements are observed (+1.1% or +0.92% R@1 improvement over Baseline). The performance is further elevated when both IPP modules are utilized concurrently, underscoring the IPP module's capacity to enrich the prototype with instance information. Fourthly, the collaborative use of DPP and IPP modules further enhances the R@1 accuracy to 74.37%, surpassing most state-of-the-art methods in Table 1. This is attributed to our Propot's ability to model instance-level and identity-level matching simultaneously. Finally, incorporating the local matching module gets the best performance for Propot.

**Impact of contextual vector length $K$ in DPP:** To facilitate the initial prototype adapt to the TIReID task, we introduce a set of learnable contextual prompt vectors for each prototype in the DPP module. The length $K$ of these vectors is a crucial parameter affecting the prototype's adaptation. To explore the impact of different vector lengths, we vary $K$ from 1 to 6 and report the results in Figure 3 (a). The observed trend indicates that larger values of

**Table 3: Performance comparison with state-of-the-art methods on RSTPReid. R@1, R@5, and R@10 are listed.**

| Methods | Pre | Ref | R@1 | R@5 | R@10 |
|---|---|---|---|---|---|
| LBUL [52] | | MM'22 | 45.55 | 68.20 | 77.85 |
| IVT [44] | | ECCVW'22 | 46.70 | 70.00 | 78.80 |
| ACSA [18] | | TMM'22 | 48.40 | 71.85 | 81.45 |
| C$_2$A$_2$ [37] | | MM'22 | 51.55 | 76.75 | 85.15 |
| PBSL [42] | w/o CLIP | MM'23 | 47.80 | 71.40 | 79.90 |
| BEAT [36] | | MM'23 | 48.10 | 73.10 | 81.30 |
| MGCN [14] | | TMM'23 | 52.95 | 75.30 | 84.04 |
| LCR$^2$S [55] | | MM'23 | 54.95 | 76.65 | 84.70 |
| TransTPS [1] | | TMM'23 | 56.05 | 78.65 | 86.75 |
| CFine [56] | | TIP'23 | 50.55 | 72.50 | 81.60 |
| VLP-TPS [48] | | arXiv'23 | 50.65 | 72.45 | 81.20 |
| IRRA [20] | | CVPR'23 | 60.20 | 81.30 | 88.20 |
| BiLMa [12] | | ICCVW'23 | 61.20 | 81.50 | 88.80 |
| DCEL [31] | w/ CLIP | MM'23 | 61.35 | 83.95 | 90.45 |
| EESSO [54] | | IVC'24 | 53.15 | 74.80 | 83.55 |
| PD [32] | | arXiv'24 | 56.65 | 77.40 | 84.70 |
| CFAM [67] | | CVPR'24 | 59.40 | 81.35 | 88.50 |
| TBPS-CLIP [2] | | AAAI'24 | 61.95 | 83.55 | 88.75 |
| **Ours** | | - | **61.87** | **83.63** | **89.70** |

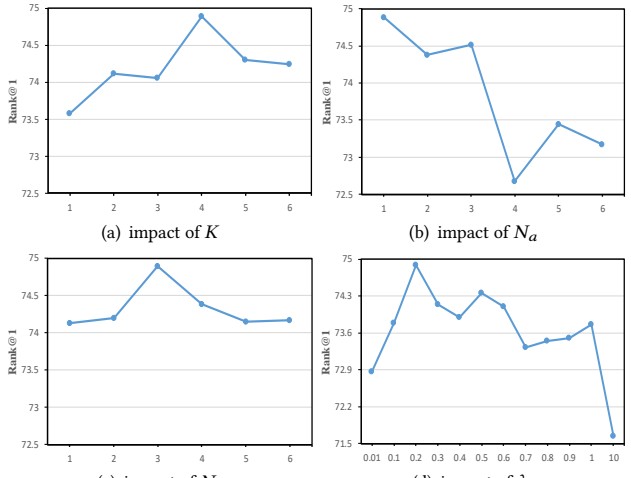

(a) impact of $K$    (b) impact of $N_a$

(c) impact of $N_e$    (d) impact of $\lambda_1$

**Figure 3: Effects of four hyper-parameters on CUHK-PEDES, including contextual vector length $K$, the block number $N_a$, $N_e$, and loss weight $\lambda_1$.**

$K$ result in better performance, as they provide more contextual parameters to capture TIReID task information. Notably, when $K$ equals 4, the performance reaches a peak of 74.89%. However, excessively large $K$ values may introduce redundant information, leading to overfitting and increased computational costs. Hence, we set $K$ to 4 to strike a balance between performance and efficiency.

**Influence of $N_a$, $N_e$:** The parameters $N_a$ and $N_e$ determine the number of blocks in SAE and CAD, respectively. Their impact on performance is shown in Figure 3 (b) and (c). Regarding $N_a$, we observe a notable performance drop when its value exceeds 3. This suggests that too many SAE parameters might impede the learning of contextual prompt vectors. Therefore, we set $N_a$ to 1 in our experiments, which yields the optimal result. Conversely, our model shows less sensitivity to $N_e$, with its curve displaying a relatively stable trend. However, excessively large $N_e$ values would introduce unnecessary parameters, leading to increased computational

**Table 4: Ablation study on different components of our Propot on CUHK-PEDES.**

| No. | Methods | IniPt | DPP | IPP Intra | IPP Inter | MLM | R@1 | R@5 | R@10 | Params | FLOPs |
|-----|---------|-------|-----|-------|-------|-----|------|------|------|--------|-------|
| 0# | Baseline | | | | | | 72.73 | 88.91 | 93.01 | 155.26M | 20.266 |
| 2# | +IniP | ✓ | | | | | 73.08 | 88.97 | 93.19 | 155.26M | 20.278 |
| 3# | +DPP | ✓ | ✓ | | | | 74.11 | 89.46 | 93.57 | 203.48M | 25.704 |
| 4# | +IPP (Intra) | ✓ | | ✓ | | | 73.83 | 89.21 | 93.53 | 158.41M | 23.058 |
| 5# | +IPP (Inter) | ✓ | | | ✓ | | 73.65 | 89.42 | 93.69 | 158.41M | 23.058 |
| 6# | +IPP | ✓ | | ✓ | ✓ | | 74.03 | 89.47 | 93.66 | 158.41M | 25.838 |
| 7# | +DPP+IPP | ✓ | ✓ | ✓ | ✓ | | 74.37 | 89.59 | 93.88 | 206.64M | 31.264 |
| 8# | +DPP+IPP+MLM | ✓ | ✓ | ✓ | ✓ | ✓ | 74.89 | 89.90 | 94.17 | 245.91M | 37.353 |

**Table 5: Ablation study of prototype aggregation schemes on CUHK-PEDES.**

| Method | Rank-1 | Rank-5 | Rank-10 |
|--------|--------|--------|---------|
| Sum | 74.30 | 89.86 | 93.84 |
| Average | 74.29 | 89.39 | 93.34 |
| MLP | 73.51 | 89.56 | 93.76 |
| Parameter | 74.19 | 89.10 | 93.18 |
| APA (Ours) | 74.89 | 89.90 | 94.17 |

costs. Hence, we set $N_e$ to 3 to achieve superior performance while maintaining efficiency.

**Impact of weight $\lambda_1$ in the objective function:** The parameter $\lambda_1$ determines the strength of identity-level matching. To assess its effect, we conduct experiments by varying $\lambda_1$ from 0.01 to 10, as depicted in Figure 3 (d). Results indicate that increasing $\lambda_1$ leads to gradual performance improvement, peaking at 0.2. However, further increase in $\lambda_1$ results in performance decline, collapsing when $\lambda_1$ reaches 10. This pattern occurs because too small $\lambda_1$ fails to effectively spread identity information to instances, making identity-level matching ineffective. Conversely, excessively large $\lambda_1$ disrupts instance-level matching by over-diffusing identity information. Therefore, we set $\lambda_1$ to 0.2 to balance instance-level and identity-level matching.

**Different Prototype Aggregation Schemes:** We introduced the Adaptive Prototype Aggregation (APA) module to adaptively combine multiple prototypes generated by different modules. To demonstrate its effectiveness, we compared APA with several common aggregation schemes: (1) summing prototypes, (2) averaging prototypes, (3) using a multi-layer perceptron (MLP) to assign weights to prototypes, and (4) learning weights for prototypes simultaneously with the network. Table 5 summarizes the comparison results. From the analysis, we draw the following conclusions: Aggregation methods with learnable weights perform worse than those without. This is because adding parameters makes optimization more difficult and uncertain for the network. Simple summation or averaging of prototypes yields superior performance, highlighting the effectiveness of our approach. The strength of our aggregation method lies in its use of the CLIP-generated initial prototype as a baseline for combining other prototypes adaptively. The quality of the initial prototype is guaranteed by CLIP's strong semantic understanding capabilities.

**Qualitative Results:** To showcase the effectiveness of Propot, we present retrieval results in Figure 4. For each query text, Figure 4 displays the top-10 gallery images retrieved by both the baseline and Propot. Baseline solely focuses on instance-level matching, whereas Propot incorporates prototype prompting to include identity-level

| Query | Top-10 Retrieval Results |
|-------|--------------------------|
| Balding male with gray hair, wearing glasses, dark colored shirt over a colored shirt, dark pants and dark colored shoes. | |
| A person with dark hair, wearing a blue short sleeve shirt, pink shorts and a pair of open heel shoes. | |
| The lady wears a black and white shirt black and white shorts with beige wedge she carries two shoulder bags black and brown. | |
| The woman is wearing a white cardigan over a long floaty dress and black shoes. She is carrying a black carry-on bag. | |

**Figure 4: Retrieval result comparisons of Baseline (the 1st row) and Propot (the 2nd row) on CUHK-PEDES. The matched and mismatched person images are marked with green and red rectangles, respectively.**

matching. The examples show that Propot excels where the baseline struggles, ensuring images with the same identity as the given query text rank high. This highlights the importance of identity-level matching, with Propot outperforming by modeling both instance-level and identity-level matching simultaneously.

## 5 CONCLUSION

In this study, to model identity-level matching for TIReID, we present Propot, a conceptually simple framework for identity-enriched prototype learning. The framework follows the 'initialize, adapt, enrich, then aggregate' pipeline. Initially, we generate robust initial prototypes using CLIP. Then, we employ the Domain-conditional Prototypical Prompting (DPP) module to prompt and adapt the initial prototypes to the TIReID task. To ensure prototype diversity, the Instance-conditional Prototypical Prompting (DPP) is devised, enriching the prototypes with both intra-modal and inter-modal instances. Finally, we use an adaptive prototype aggregation module to effectively combine multiple prototypes and diffuse their rich identity information to instances, thereby enabling identity-level matching. Through extensive experiments conducted on three popular benchmarks, we demonstrate the superiority and effectiveness of the proposed Propot framework.

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
