# OpenReview forum: "Prototypical Prompting for Text-to-image Person Re-identification"
_acmmm.org/ACMMM/2024/Conference — MM2024 Poster_

### Official Review · Reviewer_dW6n · 2024-05-15

**Rating:** 3
**Confidence:** 3

**Summary:**

This paper proposes a novel Prototypical Prompting framework (Propot) for the text-to-image person re-identification (TIReID) task, which addresses the limitations of existing methods by simultaneously modeling instance-level and identity-level matching. The key innovation lies in transforming the identity-level matching problem into an identity-enriched prototype learning problem. The proposed framework follows an 'initialize, adapt, enrich, then aggregate' pipeline. The final prototype-to-instance contrastive loss diffuses rich identity information from the prototype to each instance, enabling effective modeling of identity-level matching. Propot is designed to be single-stage and end-to-end trainable, enhancing efficiency.

**Strengths:**

1、Pioneering the use of prompt learning for solving the challenge of multi-view matching in TIReID.

2、The proposed method achieves state-of-the-art performance on multiple benchmark datasets.

**Limitations:**

1、In Table4, there is an error in the experiment serial number.

2、Q1:How are $pt_i^v$ and $pt_i^t$ initialized? Reinitialized before each epoch training or Initialized only once before model training. Q2: In the experiment Baseline+IniP, are the prototypes frozen or learnable, consistent with the DPP and IPP setup?

3、In the section "Comparisons with State-of-the-art Models", For "DECL introduces both mask language modeling and global-local semantic alignment to mine fine-grained matching, resulting in higher computational cost.".  This paper introduces an additional number of parameters to perform prototype hint learning, and there are no relevant experiments to prove that Propot has lower computational cost than DCEL.

4、In the table of comparison results with state-of-the-art models, do the bold numbers represent the case of optimal performance? It is recommended that the authors give an explanation to improve readability. For example, the bold results in Table 3 are not the highest.

5、Many state-of-the-art methods have demonstrated mAP performance, and it is recommended that the authors add this metric to their experimental results to further validate method validity.

6、Line 666-667, DECL should be DCEL. You can correct it and cite DECL: Deep evidential learning with noisy correspondence for cross-modal retrieval.

7、The SOTA approaches are not exhaustive and the paper lacks citations. For example, Noisy-Correspondence Learning for Text-to-Image Person Re-identification

**Suitability:**

3

---

### Official Review · Reviewer_8XQw · 2024-05-19

**Rating:** 5
**Confidence:** 4

**Summary:**

Based on the observation that most methods only focus on instance-level matching, this paper additionally introduces the identity-level matching, which performs the alignment between multiple images and texts of the same person. Towards this end, this paper proposes an end-to-end prototypical prompting framework (Propot) to "initialize, adapt, enrich, and then aggregate" the prototypes. Extensive experiments on three benchmarks demonstrate the effectiveness of the proposed method.

**Strengths:**

1. **Good writing**: This paper exhibits an overall clear storyline. I can easily catch the motivation and the contribution of this work.
2. **Novelty**: This paper demonstrates a high level of novelty in adapting to the specific nature of identity-level matching in TIReID, which correspondingly  presents a chain of "initialize, adapt, enrich, then aggregate" to effectively achieve identity-instance alignment.
3. **Adequate experiments**:  This paper provides a thorough experimental evaluation of the proposed method in TIReID. The authors demonstrate the effectiveness and robustness of their approach through a comprehensive set of experiments.

**Limitations:**

1. **Lack of critical reference**: This paper lacks the inclusion of several pivotal references[1] that are highly relevant to the motivation and the idea, which also delve into the identity-level matching. It is better to make a comprehensive comparison and deep analysis with the most relevant works.

2. **Inadequate Explanation**:

    (1). The methodology described in Equation 1 for generating initial prototypes needs further clarification. Traditionally, prototypes are calculated as the average values of the features of individual instances. However, Equation 1 appears to accumulate the features of each instance instead.

    (2). The dimension of $p_{a,i}$ in Equation 2 after self-attention encoder should be $(K+1) \times d$, which appears to be inconsistent with the description in Line 452.

    (3). I am having difficulty understanding the motivation behind the implementation of the cross-attention decoder, where the prototypes serve as query, while the instance features in a **batch** as key and value. It is worth considering that the instances in the batch may not encompass the entire range of identities. In cases where the prototypes lie outside the instances of the batch, how does the model handle this situation?

    (4). As the training goes on, the visual encoder and the textual encoder gradually adapt to the TIReID task. Given this adaptation process, it raises a question as to why frozen CLIP encoders are used to extract prototypes. Have the authors tried to use the on-training encoders to generate prototype features?

    Overall, the authors should provide a more detailed explanation and justification for this approach.



[1]. Bai Y, Cao M, Gao D, et al. RaSa: relation and sensitivity aware representation learning for text-based person search[C]//Proceedings of the Thirty-Second International Joint Conference on Artificial Intelligence. 2023: 555-563.

**Suitability:**

3

---

### Official Review · Reviewer_RpNk · 2024-05-23

**Rating:** 2
**Confidence:** 4

**Summary:**

This paper introduces a novel prototypical prompting framework, termed Propot, for the task of text-to-image person re-identification (TIReID). Propot addresses the identity-level matching challenge by developing identity-enriched prototypes. These prototypes are initially created using CLIP, adapted to the TIReID domain, enriched with additional instance information, and eventually aggregated into the final prototypes. The comprehensive identity information from these prototypes is then disseminated to individual instances, facilitating effective identity-level matching.

**Strengths:**

1.	The identity-level matching issue in TIReID has not been adequately resolved. This paper proposes an end-to-end trainable framework that concurrently models both instance-level and identity-level matching for TIReID.

**Limitations:**

1.	Overall, the structure of the model is somewhat intricate, and its performance does not surpass the state-of-the-art (SOTA), proving to be less than satisfactory. The comparison with SOTA methods such as [1-2] is absent.

2.	Some illustrations are unclear.

[1] 23IJCAI-Rasa: Relation and sensitivity aware representation learning for text-based person search

[2] 23MM-Towards Unified Text-based Person Retrieval: A Large-scale Multi-Attribute and Language Search Benchmark

**Suitability:**

3

---

### Meta-Review · Area_Chair_jgwG · 2024-07-07

**Recommendation:** Accept (Poster)
**Confidence:** 2

**Metareview:**

Although all reviewers have upgraded their ratings (Weak Reject->Borderline Accept, Weak Accept->Accept, Borderline Reject->Weak Accept), the following concerns still need to be addressed.
1. The idea of using CLIP for Text-to-Image retrieval is not novel.
2. The authors' feedback cannot address the concerns raised by the reviewers. In particular, "the structure of the model is somewhat intricate".
3. This paper lacks key references, which all reviewers indicated.